# Prevalence of Obstructed Defecation among Patients Who Underwent Hemorrhoidectomy and Correlation between Preoperative Constipation Score and Postoperative Patients’ Satisfaction: A Prospective Study in Two Centers

**DOI:** 10.3390/healthcare11050759

**Published:** 2023-03-05

**Authors:** Walid M. Abd El Maksoud, Khaled S. Abbas, Mohammed A. Bawahab, Fares Rayzah, Sultan M. Alkorbi, Abdulelah G. Alfaifi, Abdulrahman N. Alqahtani, Abdullah F. Alahmari, Turki B. Alotaibi

**Affiliations:** 1Surgery Department, Faculty of Medicine, King Khalid University, Abha 61421, Saudi Arabia; 2Surgery Department, Aseer Central Hospital, Abha 62523, Saudi Arabia; 3College of Medicine, King Khalid University, Abha 61421, Saudi Arabia

**Keywords:** obstructed defecation syndrome, hemorrhoidectomy, Agachan–Wexner Constipation Scoring System, postoperative patients’ satisfaction

## Abstract

Background and Objectives: obstructed defecation syndrome (ODS) is a common but underestimated condition that may affect the outcomes after hemorrhoidectomy. Therefore, the aim of this study was to determine the prevalence of obstructed defecation syndrome (ODS) among patients who underwent hemorrhoidectomy and to assess the correlation between preoperative constipation score and postoperative patients’ satisfaction. Materials and Methods: This prospective study included adult patients who underwent hemorrhoidectomy for 3rd and 4th-grade hemorrhoidal diseases. All participant patients underwent an assessment of functional OD severity by the Agachan–Wexner Constipation Scoring System. All patients were subjected to conventional hemorrhoidectomy. At 6 months postoperatively, patients were assessed again for their constipation score and postoperative patients’ satisfaction. Results: The study included 120 patients (62 males and 58 females) with a mean age of 38.7 ± 12.1 years. About one-quarter of patients (24.2%) had obstructed defecation (constipation score ≥12). ODS (constipation score ≥12) was found to be significantly more among older patients, female patients, especially those with multiple pregnancies and multiple labors, and those with perineal descent. The postoperative constipation score (5.6 ± 3.3 mean ± SD) showed significant improvement (*p* = 0.001) compared to (9.3 ± 3.9 mean ± SD) preoperatively. Postoperative patients’ satisfaction (mean 12.3 ± 3.0) at 6 months had a negative correlation with preoperative total constipation score (r = −0.035, *p* = 0.702). Conclusions: The prevalence of obstructed defecation among patients with hemorrhoids was higher than reported among the general population. High preoperative constipation scores had a negative correlation with postoperative patients’ satisfaction. Routine preoperative measurement of ODS can allow the detection of this particular group of patients who require a more physical and psychological evaluation, in addition to special preoperative counseling.

## 1. Introduction

Constipation is a gastrointestinal tract disorder that is characterized by infrequent and difficult stool passage with pain and stiffness [1]. It is a common problem that affects 2–30% of people in the Western World [2]. Constipation can occur in any age group and is more common in females [3]. A significant proportion of these patients, i.e., about 30–50% suffer from obstructed defecation syndrome [4,5]. Obstructed defecation syndrome (ODS) is a type of constipation characterized by fragmented stools, need for straining at defecation, sense of incomplete evacuation, urgency, pelvic heaviness, and self-digitation [6,7].

Hemorrhoidal disease is a very common anorectal disorder in the industrialized world [8]. Conservative therapy has shown good results in cases of 1st and 2nd-degree illness but surgical procedures are preferred in cases of 3rd and 4th-degree hemorrhoidal disease [9,10]. In patients with hemorrhoidal disease, symptoms of constipation are also often present: from the mildest such as Irritable Bowel Syndrome to the most severe such as ODS [11].

Regarding the pathophysiology of internal hemorrhoidal disease, the globally accepted theory is the sliding anal canal, or cushion theory, which postulates that abnormal slippage of cushions through the anal canal is the major pathophysiological event [12]. Constipation has traditionally been considered an important risk factor for hemorrhoid development [13]. Increased intra-abdominal pressure directly impedes venous return, via the effect of hard stool in the rectal veins, while hard stool applies strong intra-anal forces to the anal cushions during defecation [14]. Moreover, prolonged defecation attempts in the constipated patient may lead to repeated and ineffective evacuation, which impedes the venous return to hemorrhoids even more [15].

Longo who introduced a new technique called anopexis in 1993 and stapled hemorrhoidopexy for treatment of hemorrhoids postulated that hemorrhoid prolapse is always associated with an internal rectal prolapse which, and in turn, can be a symptom of ODS [16,17]. Since obstructed defecation and hemorrhoidal disease share some etiological factors that may coexist in some patients, the patients may be confused regarding their symptoms [18]. They may not be satisfied postoperatively since their undiagnosed problem of ODS may have not been solved by the hemorrhoidectomy.

Therefore, the aim of this study was to determine the prevalence of ODS among patients who underwent hemorrhoidectomy, and to assess the correlation between preoperative constipation score and postoperative patients’ satisfaction.

## 2. Materials and Methods

This is a prospective study that was conducted in Asser Central Hospital and Abha Private International Hospital, Abha, Saudi Arabia during the period from January 2021 to November 2021.

The minimal sample size was calculated according to random sample size determination equation (Dahiru et al. [19]):$$\mathrm{Sample}\;\mathrm{size}=\frac{Z^{2}*\left(p\right)*\left(1-p\right)}{c^{2}}$$
where: *Z* = *Z*-value (e.g., 1.96 for 95% confidence level), *p* = prevalence of fecal obstruction among the population: 0.08 [20,21], *c* = acceptable error, expressed as decimal. Therefore, the minimum sample size was 113.

Eligibility criteria: The study included adult patients aged 18 or more, who underwent hemorrhoidectomy as a surgical treatment for their grade III or IV hemorrhoidal diseases.

Exclusion criteria: patients aged under 18 years, with complicated hemorrhoids (such as thrombosis), or hemorrhoids combined with other anal conditions, (such as anal fissure or fistula, inflammatory bowel diseases or chronic diarrhea) were excluded from the study. In addition, patients with post-operative complications, readmission to the hospital or redo surgery were excluded from the study as these conditions could affect their satisfaction and confuse our results. Patients with previous anorectal surgery were excluded from the study.

Written informed consent was obtained from all patients regarding undergoing their operation as well as participating in the research.

### 2.1. Pre-Operative Workup

All participant patients will be assessed regarding demographic data, history taking and anthropometric measurements including weight, height, and body mass index (BMI). They underwent an assessment of functional OD severity by the Agachan–Wexner Constipation Scoring System (AWCSS) which is calculated from the answers of the patients to specific questions including how many times they go to the toilet to calculate the bowel movements [22]. Colonoscopy was used for patients with suspicious symptoms or signs or family history and those with a positive finding were excluded.

### 2.2. Operative Workup

All patients were subjected to conventional hemorrhoidectomy [23]. All the patients were operated on by the same surgical team and according to the same guideline.

### 2.3. Post-Operative Workup

Postoperative management was in accordance with the guidelines for hemorrhoidectomy with no difference regarding ODS score. Patients were discharged on the same day or the first postoperative day unless complications were anticipated. Patients were followed-up after 2, 4 and 8 weeks for detection of any complication and for assessment of the healing process.

After 6 months, all participant patients will be assessed by the Agachan–Wexner Constipation Scoring System [22] and Patient Satisfaction Score [24]. Patient satisfaction score included three questions (“Can you advise someone close to do the same surgery for the same problem?”, “With what you know now about surgery, if given the chance, would you make the same choice for the same problem?”, and “Are you satisfied with the results of surgery?”). For each question, the patient had to choose from a Likert scale (1 to 5) where 1 is very unsatisfied and 5 is very satisfied.

### 2.4. Outcomes

#### 2.4.1. Primary Endpoints

Determination of the prevalence of ODS among patients who underwent hemorrhoidectomy using the Agachan–Wexner Constipation Scoring System and considering patients who had constipation score ≥12 positive for ODS.Correlation between the preoperative constipation score using the Agachan–Wexner Constipation Scoring System and postoperative patients’ satisfaction using Patient Satisfaction Score.

#### 2.4.2. Secondary Endpoints

Determination of the correlation between the postoperative patients’ satisfaction using Patient Satisfaction Score and every element of the preoperative Agachan–Wexner Constipation Scoring System.

Data were collected, filtered then fed to a computer, using the Statistical Package for Social Sciences (IBM, SPSS, version 25,Irving, TX, USA). Descriptive statistics (frequency, percentage, mean and standard deviation) were calculated. Inferential statistics, (i.e., Chi square, *t*-test and Pearson’s correlation), were applied. *p*-values less than 0.05 was considered statistically significant.

Full confidentiality and anonymity were secured. Collected data were used only for research purposes.

## 3. Results

The study included 120 patients (62 male patients and 58 female patients) with a mean age of 38.7 ± 12.1 years. All patients were diagnosed to have symptomatic hemorrhoids (79.2% had 3rd degree hemorrhoids, while 20.8% had 4th degree hemorrhoids). The duration of symptoms ranged from 3 months to 360 months with a median of 12 months and interquartile range of 39 months, a mean of 37.2 and a standard deviation of 54.0 months. Most of the patients (82.5%) reported sensation of discomfort and only 27 patients (22.5%) had concomitant perineal descent. Three patients had previous ligation of symptomatic second-degree hemorrhoids while six patients had previous hemorrhoidectomy for a single symptomatic hemorrhoid. About one quarter of our patients (24.2%) were considered positive to have had obstructed defecation (constipation score ≥12). Correlation of items of pre-operative clinical and demographic data with the constipation score revealed that ODS (constipation score ≥12) was found to be significant in older patients. Furthermore, it was significant in female patients, especially those who had multiple pregnancies and multiple labors. Patients with preoperative perineal descent were also found to have a positive significant relation with the ODS (constipation score ≥12). Demographic and preoperative clinical data of the patients and their correlation with the preoperative AWCSS are shown in Table 1 and Table 2.

All patients were subjected to hemorrhoidectomy. The mean time of the operation was 21.3 ± 2.9 min. The operation was performed by means of cautery device in six patients (5.0%) and by means of Ligasure in 114 patients (95.0%). Most of the patients (98.3%) had their operation as a day surgery procedure. No intraoperative or postoperative complications were encountered.

The postoperative constipation score (5.6 ± 3.3 mean ± SD) showed significant improvement (*p* = 0.001) compared to (9.3 ± 3.9 mean ± SD) preoperatively. Twenty seven out of 29 patients (93.1%) who had preoperative positive ODS (constipation score ≥12) showed postoperative improvement to less than 12. However, 2 out of 91 patients (2.2%) who had negative for ODS preoperatively showed a postoperative increase and became positive. Comparison between preoperative and postoperative constipation score is shown in Table 3.

Postoperative patients’ satisfaction had a mean of 12.3 ± 3.0 at 6 months postoperatively. There was a negative correlation between postoperative satisfaction and preoperative total ODS (r = −0.035, *p* = 0.702). The more preoperative AWCSS, the lower the postoperative satisfaction. However, this relation was not significant. Preoperative abdominal pain and number of unsuccessful attempts per day showed significant negative correlation with postoperative patient satisfaction (*p* = 0.01, and 0.007 consequently). Correlation between patient’s satisfaction and the different elements of pre-operative Obstructed Defecation Score (ODS) is shown in Table 4.

Five patients were excluded from the study because of postoperative complications. Two patients were excluded because of severe postoperative pain, two patients were excluded because of delayed wound healing, and one patient was excluded because of readmission due to bleeding. There was no difference regarding complications between patients with high and low ODS scores.

## 4. Discussion

Obstructed defecation syndrome (ODS) is a common condition that affects the quality of life of the patient [25]. ODS was estimated to occur in 30–50% of all constipated patients [26]. However, the incidence of ODS among the adult population was approximately 7–8% [20,21]. We think that the high incidence of ODS among constipated patients necessitates putting it into consideration while planning for colorectal surgical conditions resulting from constipation and during the assessment of the outcomes of these conditions.

In the current study, all patients underwent hemorrhoidectomy for grade 3 or grade 4 hemorrhoids. The preoperative constipation score was more than 12 in 29 patients (24.2%). This prevalence is higher than the general population. This could be explained by the fact that people who have infrequent bowel movements and/or strain habits are more likely to suffer from hemorrhoids [27]. It has also been reported that people who strain or those who spend a long time sitting on the toilet during bowel movements are likely to develop hemorrhoids [28,29]. Nevertheless, some hemorrhoid patients reported normal bowel movements, and accordingly, it was reported that there is no sufficient evidence to make a correlation between chronic constipation and hemorrhoids [30]. From our point of view, there is no contradiction between the two opinions as constipation is not the only etiology to develop symptomatic hemorrhoids but is still considered a well-known risk factor.

The finding of the current study showed that the preoperative constipation score was significantly higher with middle and old age, female gender, increased number of pregnancies, and increased number of normal vaginal deliveries. In agreement with our finding, it was reported that ODS is typically seen in middle-aged, multiparous women. The prevalence of ODS in middle-aged women is up to 23% compared to about 3.4% in the general population [31,32,33].

Some studies suggested that ODS resulted either from a defect of pelvic support or abnormal function of the pelvic floor musculature. It was reported that childbearing damage to the innervation and soft tissues of the pelvis may have occurred as a direct consequence of vaginal childbirth [7,34]. Furthermore, it was suggested that traumatic damage to the pelvic support system does not produce immediate symptoms, which suggests that cumulative nerve damage from repeated childbirth and activities that cause chronic and repetitive increases in intraabdominal pressure play a role in the development of the symptoms. This may explain the higher prevalence of ODS in multiparous women and with older age [35,36].

In the current study, the preoperative constipation score was significantly higher in patients who had perineal descent. The patients with obstructed defecation usually have severe and prolonged straining in their trials to evacuate the rectum. This straining effort affects the pelvic floor leading to weakness and stretching of the pelvic floor muscles and ligaments. With time, this weakness leads to perineal descent. The perineal descent at first is mobile and then becomes fixed and may be associated with other pathologies like rectocele and/or rectal intussusception. When these anatomical changes happen the symptoms of obstructed defecation increase [36,37].

Wang et. al. [38] in their study concluded that the descending perineum syndrome accounts for almost 10% of tertiary referral constipation patients and is associated with concomitant rectoceles, older age, more pregnancies, more vaginal deliveries, and Ehlers-Danlos syndrome hypermobility type. Harewood et. al. [39] reported that excessive perineal descent was found in 78 % of elderly patients with evacuation disorders.

In this study, the postoperative constipation score at 6 months was improved in most of the patients who had preoperative constipation score ≥12. We do not have an explanation for this finding as hemorrhoidectomy does not treat the anatomical or functional causes of obstructed defecation. We may agree with Pescatori et. al. [40], and Dodi et. al. [41] who proposed that the placebo effect may be the explanation of this improvement. This postoperative honeymoon period, will be more obvious in very psychologically fragile patients. If we put into consideration the fact that anatomic, functional, and sometimes psychological factors are involved in defecation, this finding may be explained by psychological factors however, further studies with longer follow up are needed to investigate if this improvement will be maintained [42].

On reviewing the literature, there are very few studies that addressed the relationship between hemorrhoidectomy and constipation score. In agreement with our study, some studies reported that there was an improvement in obstructed defecation after hemorrhoidectomy, however, there were some technical differences in hemorrhoidectomy in those studies compared to the standard hemorrhoidectomy. So, improvement may also depend on the extent of the mucosal resection during hemorrhoidectomy, especially in patients that have rectoanal intussusception [43,44].

In the current study, the more preoperative ODS, the lower the postoperative satisfaction. However, this relation was not significant. Some of ODS items such as preoperative abdominal pain and number of unsuccessful attempts per day showed significant negative correlation with postoperative patient satisfaction. Although OD is a risk factor for hemorrhoids, we could not find any study addressing the effect of high ODS on patient satisfaction after hemorrhoidectomy.

The limitations of the current study included short term follow-up, and the patient satisfaction score used was not specific to anal surgery.

## 5. Conclusions

The prevalence of obstructed defecation among patients with hemorrhoids was higher than what was reported among the general population. High preoperative constipation scores had a negative correlation with postoperative patients’ satisfaction. Routine preoperative measurement of ODS can allow the detection of this particular group of patients who require a more physical and psychological evaluation, in addition to special preoperative counseling.

## 6. Recommendation

More studies with a larger number of patients and longer follow up are needed to establish new clear guidelines that necessitate evaluation of OSD using simple method as constipation score before management of hemorrhoids.

## Figures and Tables

**Table 1 healthcare-11-00759-t001:** Demographic and clinical data of the studied group (N = 120).

	Number (120)	%
Age		
• <30 years	36	30.0
• 30–39 years	29	24.2
• 40–49 years	35	29.2
• >50 years	20	16.7
Mean ± SD	38.7 ± 12.1 years	
Gender		
• Males	62	51.7
• Females	58	48.3
BMI		
• Normal	44	36.7
• Overweight	52	43.3
• Obese	24	20.0
Mean ± SD	26.7 ± 5.1 years	
Degree of the hemorrhoids		
• Third degree	95	79.2
• Fourth degree	25	20.8
Existing concomitant perineal descent		
• No perineal descent	93	77.5
• There is perineal descent	27	22.5
Duration of symptoms		
• < 1 year	47	39.2
• 1–3 years	43	35.8
• >3 years	30	25.0
Number of previous pregnancies (*n* = 58)		
• 0	15	25.9
• 1–4	23	39.7
• 5+	20	34.5
Number of normal labors (*n* = 58)		
• 0	26	44.8
• 1–4	12	20.7
• 5+	20	34.5
Previous anal procedure and/or surgeries		
• No	111	92.5
• Yes	9	7.5
Previous abdominal surgeries		
• 0	84	70.0
• 1	34	28.3
• 2	2	1.7
Preoperative constipation score		
• No obstructed defecation (<12)	91	75.8
• Positive for obstructed defecation (≥12)	29	24.2

**Table 2 healthcare-11-00759-t002:** Correlation of every item of pre-operative clinical and demographic data with the Agachan–Wexner Constipation Scoring System.

Variable			Agachan–Wexner Constipation Scoring System			*p*-Value
			Score < 12	Score ≥ 12	Total	
Age	<30 years	Count	30	6	36	
		%	83.3%	16.7%	100.0%	
	30–39 years	Count	24	5	29	
		%	82.8%	17.2%	100.0%	
	40–49 years	Count	29	6	35	
		%	82.9%	17.1%	100.0%	
	50+ years	Count	8	12	20	
		%	40.0%	60.0%	100.0%	
	Total	Count	91	29	120	
		%	75.8%	24.2%	100.0%	0.001 *
Gender	Male	Count	52	10	62	
		%	83.9%	16.1%	100.0%	
	Female	Count	39	19	58	
		%	67.2%	32.8%	100.0%	
	Total	Count	91	29	120	
		%	75.8%	24.2%	100.0%	0.033 *
Preoperative BMI	Normal	Count	32	12	44	
		%	72.7%	27.3%	100.0%	
	Overweight	Count	42	10	52	
		%	80.8%	19.2%	100.0%	
	Obese	Count	17	7	24	
		%	70.8%	29.2%	100.0%	
	Total	Count	91	29	120	
		%	75.8%	24.2%	100.0%	0.535
Duration of Symptoms	< 1 year	Count	31	16	47	
		%	66.0%	34.0%	100.0%	
	1–3 years	Count	37	6	43	
		%	86.0%	14.0%	100.0%	
	>3 years	Count	23	7	30	
		%	76.7%	23.3%	100.0%	
	Total	Count	91	29	120	
		%	75.8%	24.2%	100.0%	0.084
Degree of piles	3rd degree	Count	71	24	95	
		%	74.7%	25.3%	100.0%	
	4th degree	Count	20	5	25	
		%	80.0%	20.0%	100.0%	
	Total	Count	91	29	120	
		%	75.8%	24.2%	100.0%	0.584
Previous pregnancies	0	Count	13	2	15	
		%	86.7%	13.3%	100.0%	
	1–4	Count	20	3	23	
		%	87.0%	13.0%	100.0%	
	5+	Count	10	10	20	
		%	50.0%	50.0%	100.0%	
	Total	Count	43	15	58	
		%	74.1%	25.9%	100.0%	0.010 *
Previous deliveries	0	Count	22	4	26	
		%	84.6%	15.4%	100.0%	
	1–4	Count	11	1	12	
		%	91.7%	8.3%	100.0%	
	5+	Count	10	10	20	
		%	50.0%	50.0%	100.0%	
	Total	Count	43	15	59	
		%	74.1%	25.9%	100.0%	0.007 *
Previous anal surgeries	No	Count	87	24	111	
		%	78.4%	21.6%	100.0%	
	Yes	Count	4	5	9	
		%	44.4%	55.6%	100.0%	
	Total	Count	91	29	120	
		%	75.8%	24.2%	100.0%	0.022
Previous abdominal surgeries	0	Count	66	18	84	
		%	78.6%	21.4%	100.0%	
	1	Count	23	11	34	
		%	67.6%	32.4%	100.0%	
	2	Count	2	0	2	
		%	100.0%	0.0%	100.0%	
	Total	Count	91	29	120	
		%	75.8%	24.2%	100.0%	0.329
Perineal descent	No	Count	76	17	93	
		%	81.7%	18.3%	100.0%	
	Yes	Count	15	12	27	
		%	55.6%	44.4%	100.0%	
	Total	Count	91	29	120	
		%	75.8%	24.2%	100.0%	0.005 *

* *p* < 0.05 is significant.

**Table 3 healthcare-11-00759-t003:** Comparison between pre and post operative constipation scores (Agachan–Wexner Constipation Scoring System).

			Postoperative Grade			*p* Value
			**<12**	**≥12**	**Total**	
Preoperative Grade	<12	Count	89	2	91	
		%	97.8%	2.2%	100.0%	
	>12	Count	27	2	29	
		%	93.1%	6.9%	100.0%	
	Total	Count	116	4	120	
		%	96.7%	3.3%	100.0%	0.220

**Table 4 healthcare-11-00759-t004:** Correlation between patient’s satisfaction and the different elements of pre-operative constipation scores (Agachan–Wexner Constipation Scoring System).

Elements of the Preoperative Agachan–Wexner Constipation Scoring System		Satisfaction	
Frequency of bowel movement per day		Pearson Correlation	0.068
		Sig. (2-tailed)	0.459
Painful evacuation effort during defecation		Pearson Correlation	0.072
		Sig. (2-tailed)	0.435
Feeling of incomplete evacuation		Pearson Correlation	−0.023
		Sig. (2-tailed)	0.801
Abdominal pain		Pearson Correlation	−0.234 *
		Sig. (2-tailed)	0.010 *
Minutes in lavatory per attempt		Pearson Correlation	−0.008
		Sig. (2-tailed)	0.933
Type of assistance		Pearson Correlation	−0.094
		Sig. (2-tailed)	0.309
Unsuccessful attempt per day		Pearson Correlation	−0.244 *
		Sig. (2-tailed)	0.007 *
Duration of constipation (years)		Pearson Correlation	−0.062
		Sig. (2-tailed)	0.499
		Total Number	120

* *p* < 0.05 is significant.

## Data Availability

Data will be available from the corresponding author and will be released on reasonable request.

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
