# Peer review of "Prevalence of Obstructed Defecation among Patients Who Underwent Hemorrhoidectomy and Correlation between Preoperative Constipation Score and Postoperative Patients’ Satisfaction: A Prospective Study in Two Centers"

_healthcare, 2023, doi:10.3390/healthcare11050759_

Round 1

Reviewer 1 Report

This is a prospective article on the study of the correlation between constipation and intestinal hemorrhoid surgery. This article is very interesting and has a certain guiding significance for the evaluation of clinical treatment and effects after being banned.

1In Table 1, we can see some specific data. The number of overweight patients is relatively large. The author should give a corresponding explanation and analysis in the discussion part.

2In Table 2, we can see that the percentage of male ratings is significantly higher than that of female ratings in Score <12, however, in Score ≥ 12The number and rating of women is significantly higher than that of men, which should be explained in the discussion.

3Table 3. Comparison between pre and post-operative constipation scores However, in Table 3, we do not see the content of postoperative constipation evaluation. Please pay attention to whether the statement in the table is incorrect.

4In Table 4, there is an evaluation of the frequency of bowel movements per day. How to evaluate this in clinical practice and what methods are used to record it needs to be explained in the article.

5The authors mentioned in the discussion that there was significant relief of symptoms 6 months after hemorrhoid surgery after constipation, but there was no proper reason to discriminate against this phenomenon. The author mentioned that the use of a placebo may explain this reason, but for the vast majority of people, there is no use of a placebo, and the symptoms are relieved. We believe that there are more reasonable anatomical or psychological explanations, rather than a placebo. Please further optimize the content of this part of the discussion.

6In the conclusion part, it is mentioned that it is more necessary to use spiritual or psychological comfort for the difficulty of constipation and postoperative. For patients with constipation and hemorrhoids, if we want to alleviate this pain, what we should get in this article is that if the constipation score is higher, the postoperative relief should be more obvious, so we should mention early prevention and early treatment in the conclusion part, which is of great significance for alleviating constipation and hemorrhoids.

Author Response

Dear Chief Editor

          We would like to thank you and the editorial board for the tremendous efforts you are doing in revising and processing our manuscript.

          We also appreciate the valuable comments of the reviewers that raised important points which will improve the quality of the manuscript. The following are our point-by-point responses to the reviewers’ comments:

Reviewer 1:

  • In Table 1, we can see some specific data. The number of overweight patients is relatively large. The author should give a corresponding explanation and analysis in the discussion part.
    • The prevalence of obesity in Saudi Arabia was found to be high for a long time. Studies conducted from 1990 to 1993, have shown an overall prevalence of obesity of 22.1% and approximately 53% of Saudi adults are either overweight or obese
    • Other studies in 2005 found that the prevalence of overweight was 36.9%. Being overweight is significantly more prevalent in males (42.4%) compared to 31.8% of females (p<0.0001). The age-adjusted prevalence of obesity was 35.5% in KSA with an overall prevalence of 35.6% [95% CI: 34.9-36.3], while severe (gross) obesity was 3.2%. Females are significantly more obese with a prevalence of 44% than males 26.4% (p<0.0001).
    • It was found that the weighted prevalence of obesity in KSA was 25.6% in 2018 and 28.7% in 2013 and 24.7% in 2020
    • So, we found that the results of our study are within the range of obesity reported in KSA
    • References:
      • Al-Nuaim AA, Bamgboye EA, al-Rubeaan KA, al-Mazrou Y. Overweight and obesity in Saudi Arabian adult population, the role of socio-demographic variables. J Community Health 1997; 3: 211-223.
      • Al-Nozha, Mansour & Al-Mazrou, Yagob & Al-Maatouq, Mohammed & Arafah, Mohammed & Khalil, Mohamed & Khan, Nazeer & Al-Marzouki, Khalid & Abdullah, Moheeb & Al-Khadra, Akram & Al-Harthi, Saad & Al Shahid, Maie & Al-Mobeireek, Abdulellah & Nouh, Mohammed. (2005). Obesity in Saudi Arabia. Saudi medical journal. 26. 824-829.
      • Althumiri NA, Basyouni MH, AlMousa N, AlJuwaysim MF, Almubark RA, BinDhim NF, Alkhamaali Z, Alqahtani SA. Obesity in Saudi Arabia in 2020: Prevalence, Distribution, and Its Current Association with Various Health Conditions. Healthcare (Basel). 2021 Mar 11;9(3):311. doi: 10.3390/healthcare9030311. PMID: 33799725; PMCID: PMC7999834.

  • In Table 2, we can see that the percentage of male ratings is significantly higher than that of female ratings in Score <12, however, in Score ≥ 12The number and rating of women are significantly higher than that of men, which should be explained in the discussion.
    • These findings were discussed in the 3rd and 4th paragraphs in the discussion section as follows.
      • The finding of the current study showed that the preoperative constipation score was significantly higher with middle and old age, female gender, increased number of pregnancies, and increased number of normal vaginal deliveries. In agreement with our finding, it was reported that ODS is typically seen in middle-aged, multiparous women. The prevalence of ODS in middle-aged women is up to 23% compared to about 3.4% in the general population.
      • Some studies suggested that ODS resulted from either a defect of pelvic support or abnormal function of the pelvic floor musculature. It was reported that childbearing women’s damage to the innervation and soft tissues of the pelvis may have occurred as a direct consequence of vaginal childbirth (34, 35). Furthermore, it was suggested that traumatic damage to the pelvic support system does not produce immediate symptoms, which suggests that cumulative nerve damage from repeated childbirth and activities that cause chronic and repetitive increases in intraabdominal pressure play a role in the development of the symptoms. This may explain the higher prevalence of ODS in multiparous women and with older age.
      • References:
        1. Gaspari AL, Sileri P. Pelvic floor disorders: surgical approach: Springer; 2014.
        2. Snooks S, Swash M, Setchell M, Henry M. Injury to innervation of pelvic floor sphincter musculature in childbirth. The Lancet. 1984;324(8402):546-50.
        3. Stewart WF, Liberman JN, Sandler RS, Woods MS, Stemhagen A, Chee E, et al. Epidemiology of constipation (EPOC) study in the United States: relation of clinical subtypes to sociodemographic features. The American journal of gastroenterology. 1999;94(12):3530-40.
        4. Ellis CN, Essani R. Treatment of obstructed defecation. Clinics in Colon and Rectal Surgery. 2012;25(01):024-33.
        5. Sultan A, Monga A, Stanton S. The pelvic floor sequelae of childbirth. British journal of hospital medicine. 1996;55(9):575-9.
        6. Gill EJ, Hurt WG. Pathophysiology of pelvic organ prolapse. Obstetrics and gynecology clinics of North America. 1998;25(4):757-69.

  • Comparison between pre and post-operative constipation scores However, in Table 3, we do not see the content of postoperative constipation evaluation. Please pay attention to whether the statement in the table is incorrect.
    • We thank the reviewer for his comment. It was corrected in table 3 to be (postoperative).

  • In Table 4, there is an evaluation of the frequency of bowel movements per day. How to evaluate this in clinical practice and what methods are used to record it needs to be explained in the article.
    • This point was clarified in the preoperative work-up section, first paragraph, line 105 as per the reviewer's suggestion.

  • The authors mentioned in the discussion that there was significant relief of symptoms 6 months after hemorrhoid surgery after constipation, but there was no proper reason to discriminate against this phenomenon. The author mentioned that the use of a placebo may explain this reason, but for the vast majority of people, there is no use of a placebo, and the symptoms are relieved. We believe that there are more reasonable anatomical or psychological explanations, rather than a placebo. Please further optimize the content of this part of the discussion.
    • We agree with the reviewer that placebo may not be the cause for the improvement although suggested by some authors as mentioned in the discussion section, 7th We think that the improvement may be due to psychological factors, however, further studies with longer follow-up are needed to figure out if this improvement will be maintained. This was added to the discussion section, the 7th paragraph, line 245.
    • On reviewing the literature, there are very scanty studies that addressed the relationship between hemorrhoidectomy and constipation score. In agreement with our study, some studies reported that there was an improvement in obstructed defecation after hemorrhoidectomy, however, there were some technical differences in hemorrhoidectomy in those studies compared to the standard hemorrhoidectomy. So, Improvement may also depend on the extent of the mucosal resection during hemorrhoidectomy, especially in patients that have rectoanal intussusception. This was added to the discussion section, paragraph 8, line 255.

  • In the conclusion part, it is mentioned that it is more necessary to use spiritual or psychological comfort for the difficulty of constipation and postoperative. For patients with constipation and hemorrhoids, if we want to alleviate this pain, what we should get in this article is that if the constipation score is higher, the postoperative relief should be more obvious, so we should mention early prevention and early treatment in the conclusion part, which is of great significance for alleviating constipation and hemorrhoids.
    • We agree with the reviewer, that this is very important that those patients with preoperative high scores should be counseled and treated differently.
    • Accordingly, we concluded that these patients should be identified preoperatively using the constipation score and should have a more physical and psychological evaluation for assessment of their condition.
    • Furthermore, we added a recommendation “More studies with a larger number of patients and longer follow up is needed to establish new clear guidelines that necessitate an evaluation of OSD using a simple method as constipation score before management of hemorrhoids.” In a recommendation section, line 274.

Reviewer 2 Report

Several issues need to be addressed:

1. Were all patients operated on by the same surgical team and according to the same guideline?

2. Were colonoscopy used for selected patients?

3. Most important of all, why did the authors not prefer to treat coexisting pelvic organ prolapse and/or ODS due to rectocele before or simultaneously with the hemorrhoidal disease? We know that patients with ODS frequently have anal disorders as the "tip of the iceberg". This concept should be well-emphasized. Otherwise, the deduction will lose power, and the reader maight think that we can simply neglect an underlying ODS/POP and simply go on with the surgicall treatment of any advanced hemorrhoidal disease

Author Response

Dear Chief Editor

          We would like to thank you and the editorial board for the tremendous efforts you are doing in revising and processing our manuscript.

          We also appreciate the valuable comments of the reviewers that raised important points which will improve the quality of the manuscript. The following are our point-by-point responses to the reviewers’ comments:

Reviewer 2:

  • Were all patients operated on by the same surgical team and according to the same guideline?
    • Yes, all the patients were operated on by the same surgical team and according to the same guideline This was added to the Operative Workup section, line 112.
  • Was colonoscopy used for selected patients?
    • It is a really important point to be mentioned, we thank the reviewer for his comment.
    • Yes, colonoscopy was used for patients with suspicious symptoms or signs or family history, and those with positive findings were excluded. This was added to the Preoperative Workup section, line 108.

  • Most important of all, why did the authors not prefer to treat coexisting pelvic organ prolapse and/or ODS due to rectocele before or simultaneously with the hemorrhoidal disease? We know that patients with ODS frequently have anal disorders as the "tip of the iceberg". This concept should be well emphasized. Otherwise, the deduction will lose power, and the reader might think that we can simply neglect an underlying ODS/POP and simply go on with the surgical treatment of any advanced hemorrhoidal disease
    • This is a very important point. We agree that hemorrhoids may be the tip of the iceberg so we emphasized in our conclusion on routine use of constipation scores during the evaluation of patients with hemorrhoids to detect patients with ODS that may need more evaluation and different management.
    • There are no available guidelines that necessitate an evaluation of OSD before the management of hemorrhoids. We conducted this study as a step to prove that we need new clear guidelines that necessitate an evaluation of OSD before the management of minor anal conditions and of course, this will need more studies with a larger number of patients and longer follow-up.
    • In addition to our conclusions, we added recommendations (recommendation section, line 275) to stress the importance of having special guidelines to detect patients with ODS among patients who are going to undergo hemorrhoidectomy as they need special care.

Reviewer 3 Report

In my opinion the study design is not appropriate, the population is not homogeneous. Infact  One -quarter of the included patients had an ODS but is not reported if they were studied with colonoscopy, defecography and or anorectal manometry in order to investigate other clinical  conditions related to the ODS, such as: rectocele, anorectal intussusception, pelvic floor dyssynergia, enterocele etc.. This evaluation is fundamental to reduce methologic errors.

Therefore these patients were yet in therapy with stool softner?

It' not clear if t he exclusion criteria included also any pathology and/or therapy that may cause ODS.

It's not reported the number of patients excluded from the study because of readmission or redo surgery. Are the complications  more frequent in patients with ODS?

It's also important to describe the postoperative  clinical management : 

diet, laxative, analgesia with oppioids or not.

In my opinionthe  results and the discussion are not conclusive and have not a strong scientific significance.

Author Response

Dear Chief Editor

          We would like to thank you and the editorial board for the tremendous efforts you are doing in revising and processing our manuscript.

          We also appreciate the valuable comments of the reviewers that raised important points which will improve the quality of the manuscript. The following are our point-by-point responses to the reviewers’ comments:

Reviewer 3:

  • In my opinion, the study design is not appropriate, the population is not homogeneous. In fact, One - a quarter of the included patients had an ODS but is not reported if they were studied with colonoscopy, defecography, and or anorectal manometry in order to investigate other clinical conditions related to the ODS, such as rectocele, anorectal intussusception, pelvic floor dyssynergia, enterocele, etc. This evaluation is fundamental to reducing methodologic errors. Therefore, these patients were yet in therapy with stool softener?
    • We agree that every patient with hemorrhoid should be evaluated for ODS before planning for the management and highlighting this point was an aim of this study. Although Hemorrhoidal disease is a common disorder requiring surgical intervention in approximately 10% of cases, there are no clear guidelines that necessitate an evaluation of OSD before the management of hemorrhoids. In addition, the large number of patients undergoing hemorrhoidectomy is an obstacle to applying all sophisticated investigations for all of them. Hence the importance of applying a selective approach.
    • In our study, we tried to find a cheap easy mothed constipation score to detect the hidden ODS complaints among patients who were going to have hemorrhoidectomy because of severe symptoms of hemorrhoids and to study the effect of high scores on the postoperative patients’ satisfaction.
    • Results of the current study showed that a high preoperative ODS score may lead to less postoperative patient satisfaction. So, we emphasized in the discussion section and conclusions on the routine use of constipation score during the evaluation of patients with hemorrhoids to detect the patients that need detailed investigation before planning for management    

  • It's  not clear if the exclusion criteria included also any pathology and/or therapy that may cause ODS.
    • It is really important to clarify this point. All Patients with previous anorectal surgery were excluded from the study. Add to methodology. This was added to the Materials and Method section, Exclusion Criteria, line 98.

  • It's not reported the number of patients excluded from the study because of readmission or redo surgery. Are the complications more frequent in patients with ODS?
    • Five patients were excluded from the study because of postoperative complications. Two patients were excluded because of severe postoperative pain, 2 patients were excluded because of delayed wound healing, and one patient was excluded because of readmission due to bleeding. There was no difference regarding complications between patients with high and low ODS scores.
    • This was added to the Results section, 5th paragraph, line 184.

  • It's also important to describe the postoperative clinical management: diet, laxative, analgesia with opioids or not.
    • Postoperative management was according to the guideline for hemorrhoidectomy with no difference regarding ODS score.
    • This was added to the Postoperative Workup section, line 117 as per the reviewer's suggestion.